# Rapid Dynamic Naturalistic Monitoring of Bradykinesia in Parkinson’s Disease Using a Wrist-Worn Accelerometer

**DOI:** 10.3390/s21237876

**Published:** 2021-11-26

**Authors:** Jeroen G. V. Habets, Christian Herff, Pieter L. Kubben, Mark L. Kuijf, Yasin Temel, Luc J. W. Evers, Bastiaan R. Bloem, Philip A. Starr, Ro’ee Gilron, Simon Little

**Affiliations:** 1Department of Neurosurgery, School of Mental Health and Neuroscience, Maastricht University, 6229 ER Maastricht, The Netherlands; c.herff@maastrichtuniversity.nl (C.H.); p.kubben@maastrichtuniversity.nl (P.L.K.); y.temel@maastrichtuniversity.nl (Y.T.); 2Department of Neurology, School of Mental Health and Neuroscience, Maastricht University, 6229 ER Maastricht, The Netherlands; mark.kuijf@mumc.nl; 3Department of Neurology, Center of Expertise for Parkinson & Movement Disorders, Donders Institute for Brain, Cognition and Behaviour, Radboud University Medical Center, 6525 GC Nijmegen, The Netherlands; Luc.Evers@radboudumc.nl (L.J.W.E.); bas.bloem@radboudumc.nl (B.R.B.); 4Department of Movement Disorders and Neuromodulation, University of California San Francisco, San Francisco, CA 94143, USA; philip.starr@ucsf.edu (P.A.S.); roeegilron@gmail.com (R.G.); Simon.Little@ucsf.edu (S.L.)

**Keywords:** Parkinson’s disease, bradykinesia, real-life, naturalistic monitoring, wearable sensors, accelerometer, motor fluctuation

## Abstract

Motor fluctuations in Parkinson’s disease are characterized by unpredictability in the timing and duration of dopaminergic therapeutic benefits on symptoms, including bradykinesia and rigidity. These fluctuations significantly impair the quality of life of many Parkinson’s patients. However, current clinical evaluation tools are not designed for the continuous, naturalistic (real-world) symptom monitoring needed to optimize clinical therapy to treat fluctuations. Although commercially available wearable motor monitoring, used over multiple days, can augment neurological decision making, the feasibility of rapid and dynamic detection of motor fluctuations is unclear. So far, applied wearable monitoring algorithms are trained on group data. In this study, we investigated the influence of individual model training on short timescale classification of naturalistic bradykinesia fluctuations in Parkinson’s patients using a single-wrist accelerometer. As part of the Parkinson@Home study protocol, 20 Parkinson patients were recorded with bilateral wrist accelerometers for a one hour OFF medication session and a one hour ON medication session during unconstrained activities in their own homes. Kinematic metrics were extracted from the accelerometer data from the bodyside with the largest unilateral bradykinesia fluctuations across medication states. The kinematic accelerometer features were compared over the 1 h duration of recording, and medication-state classification analyses were performed on 1 min segments of data. Then, we analyzed the influence of individual versus group model training, data window length, and total number of training patients included in group model training, on classification. Statistically significant areas under the curves (AUCs) for medication induced bradykinesia fluctuation classification were seen in 85% of the Parkinson patients at the single minute timescale using the group models. Individually trained models performed at the same level as the group trained models (mean AUC both 0.70, standard deviation respectively 0.18 and 0.10) despite the small individual training dataset. AUCs of the group models improved as the length of the feature windows was increased to 300 s, and with additional training patient datasets. We were able to show that medication-induced fluctuations in bradykinesia can be classified using wrist-worn accelerometry at the time scale of a single minute. Rapid, naturalistic Parkinson motor monitoring has the clinical potential to evaluate dynamic symptomatic and therapeutic fluctuations and help tailor treatments on a fast timescale.

## 1. Introduction

Parkinson’s disease (PD) is a disabling neurodegenerative disorder characterized by motor and non-motor symptoms that affect patients’ motor performance and quality of life (QoL) [1,2,3]. Symptomatic PD management often initially focuses on dopamine replacement therapies [4]. However, half of PD patients develop wearing-off motor fluctuations during the first decade after diagnosis [5,6]. Wearing-off motor fluctuations are defined as inconsistent therapeutic benefits on symptoms such as bradykinesia and rigidity, despite regular dopaminergic delivery [6]. These motor fluctuations and other dopaminergic-related side effects can markedly impair patients’ QoL [7]. Motor fluctuations are therefore a primary indication for consideration of deep brain stimulation (DBS) [1,8]. Adequate monitoring of motor fluctuations is essential for treatment evaluation, both in the presence and absence of DBS, and wearable motion sensing represents an appealing approach to support this quantification [9,10], although several challenges remain to be addressed [11,12].

Ideally, objective motor fluctuation monitoring should accurately measure and decode movement, during real world (naturalistic) activities, and be simple to implement for patients [10,13]. Currently used Parkinson’s evaluation tools, such as the Movement Disorders Society Unified Parkinson Disease Rating Scale (MDS-UPDRS) and the Parkinson Disease QoL questionnaire (PDQ-39), are not designed for chronic dynamic naturalistic symptom monitoring [14,15]. They contain questionnaires which capture subjective estimates of retrospective symptoms over a week (MDS-UPDRS II and IV) or a month (PDQ-39). They are also dependent on patient recall, which is often imperfect, particularly in patients with cognitive dysfunction. Observing and scoring motor fluctuations requires trained health providers to perform single time point evaluations (MDS-UPDRS III). Motor diaries, often used as the gold standard for 24-h naturalistic monitoring, require self-reporting every 30 min [16]. This burden causes recall bias and diary fatigue over long-term use [17].

The strong clinical need for continuous symptom tracking, together with the wide availability and presence of affordable accelerometer-based sensors, has led to several academic and commercially available wearable sensor PD monitoring systems [18,19,20,21,22]. Motor fluctuation monitoring with commercially available devices is currently based primarily on summary metrics derived from multiple days of sensor data. Incorporation of these metrics during neurological consultation has led to promising augmentation of clinical decision making [21,23,24,25,26]. However, these sensor monitoring systems have thus far been found to be better correlated with PD clinical metrics on a time scale of days rather than hours [21,27], which is a notably longer time window than used in the original development studies [19,21,28,29,30]. Successful motor fluctuation classification over shorter time periods (minutes or hours) would enable dynamic therapeutic motor response monitoring. Thus, we have suggested individual model training as a methodological improvement to pursue this. To date, motor monitoring algorithms have typically been trained on group data; individual model training is suggested due to inter-subject heterogeneity of PD symptomatology [11,18,20,31]. This hypothesis is strengthened by a recent successful algorithm innovation combining short- and long-time epochs in a deep learning model correlating wrist and ankle-worn accelerometer metrics with total UPDRS III scores on 5 min epochs [32].

In the present work, we investigated the performance of machine learning classification models identifying rapid (single minute resolution), medication-induced motor fluctuations in PD patients. The classification models were trained on unconstrained naturalistic (at home) motion data derived from a unilateral wrist-worn accelerometer. Classification models based on individual data were compared with models based on group data. Further, we analyzed the influence of the number of individuals included in the group model training data, and the length of analyzed accelerometer data epochs (time window lengths), on classification results. We focused symptom decoding on bradykinesia since this cardinal feature of PD has been found to be more challenging to detect with motion sensors than tremor or dyskinesia [1,33]. This is likely due to higher distributional kinematic overlap of bradykinesia fluctuations with normal movements and normal periods of rest [18,34,35,36].

We hypothesized that single-minute bradykinesia classification would be achievable using machine learning and that individualized motion classification models in PD would demonstrate more reliable short-term classification of naturalistic bradykinesia fluctuations compared to group models.

## 2. Material and Methods

### 2.1. Study Sample

For our analysis, we used data from 20 participants of the Parkinson@Home validation study [37]. Detailed descriptions of the study’s protocol and feasibility have been described previously [38,39]. In brief, the Parkinson@Home study recruited 25 patients diagnosed with PD by a movement disorders neurologist. All patients underwent dopaminergic replacement treatment with oral levodopa therapy, experienced wearing-off periods (MDS-UPDRS part IV item 4.3 ≥ 1) and had at least slight Parkinson-related gait impairments (MDS-UPDRS part II item 2.12 ≥ 1 and/or item 2.13 ≥ 1). Participants who were treated with advanced therapies (DBS or infusion therapies) or who suffered significant psychiatric or cognitive impairments which hindered completion of the study protocol were excluded.

For the current subset of PD patients, we included 20 patients who showed a levodopa-induced improvement in unilateral upper extremity bradykinesia at least at one side (equal or less than zero points). Unilateral upper extremity bradykinesia was defined as the sum of MDS-UPDRS part III items 3c, 4b, 5b, and 6b for the left side, and items 3b, 4a, 5a, and 6a for the right side. Sum scores from medication on-states were compared with sum scores from medication off-states. For each included participant, only data from the side with the largest clinical change in upper extremity bradykinesia sub items were included. Patients’ activities were recorded on video and annotated as described in the validation study [37].

For our current analysis, only unilateral wrist tri-axial accelerometer data from the side with the largest fluctuation in unilateral upper extremity bradykinesia were included (Gait Up Physilog 4, Gait Up SA, CH). Recordings consisted of two sessions which took place on the same day. First, the pre-medication recording was performed in the morning after overnight withdrawal of dopaminergic medication. Second, the post-medication recording was performed when the participants experienced the full clinical effect after intake of their regular dopaminergic medication. During both recordings, participants performed an hour of unconstrained activities within and around their houses. At the start of both recordings, a formal MDS-UPDRS III and Abnormal Involuntary Movement Scale (AIMS) was conducted by a trained clinician.

### 2.2. Data Pre-Processing and Feature Extraction

Accelerometer data were sampled at 200 Hz and downsampled to a uniform sampling rate of 120 Hertz (Hz) using piecewise cubic interpolation. The effect of gravity was removed from each of the three time series (x-, y-, and z-axes) separately, by applying an l1-trend filter designed to analyze time series with an underlying piecewise linear trend [40]. Time series were low-pass filtered at 3.5 Hz to attenuate frequencies typically associated with Parkinsonian tremor in accelerometer time series [41]. In addition to the three individual accelerometer time series, we computed a composite time series containing the vector magnitude of the three individual accelerometer axes (x^2^ + y^2^ + z^2^).

Multiple features previously shown to correlate with bradykinesia were extracted from the four time series (in total 103 features from four time series: x, y, z, and vector magnitude) (see extensive overview including references in Appendix A). The features included characteristics from the temporal domain (such as extreme values, variances, jerkiness, number of peaks, and root mean squares), the spectral domain (such as spectral power in specific frequency ranges), and dominant frequencies. The standard window length of analysis for each extracted feature was set at 60 s, meaning one mean value per feature was extracted per time series over every 60 s of data. To explore the influence of varying window lengths (3, 10, 30, 90, 120, 150, and 300 s), separate feature sets were extracted for each sub analysis. All individual feature sets were balanced for medication-status by discarding the surplus of available data in the longest recording (pre- or post-medication). Features were standardized by calculating individual z-scores per feature. To not average out pre- and post-medication differences, the mean of only the pre-medication recordings was extracted from a value, and the result was divided by the standard deviation of only the pre-medication recordings [42]. To test the influence of an activity filter, data windows without motion activity were identified. For this, different methodologies of activity filtering are described in PD monitoring literature [9,43,44]. We applied an activity filter which classified every 60 s window with a coefficient of variation of the vector magnitude less than 0.3 as no activity and discarded them from the analysis. The choice of selected feature was based on previous work [43], and the threshold was chosen pragmatically by group-level observations of video-annotated sections identified as non-active [37]. The activity-filtered data sets were individually balanced for medication-states. For example, if a participant’s data set resulted in 50 active minutes of pre-medication data, and only 45 active minutes of post-medication data, the surplus of features from 5 active minutes of pre-medication data was discarded at the end of the data set.

### 2.3. Descriptive Statistics and Analysis of Variance

The demographic and disease characteristics of the included participants are described in Table 1. Unilateral scores are provided only for the side on which accelerometer data was analyzed.

To test the statistical distinguishability of the pre- and post-medication recordings at the group level before using the entire dataset as an input, four main accelerometer features were chosen a priori. These four features covered the most often used domains of motion metrics applied for naturalistic bradykinesia monitoring (maximum acceleration, coefficient of variation of acceleration over time, root mean square of acceleration over time, and the total spectral power below 4 Hz) [18,43] and were extracted from the vector magnitude time series. Individual averages of each of the four features over the entire dataset (~60 min per condition) were analyzed for statistically significant differences between the medication states with a multivariate analysis of variance (M-ANOVA). Post-hoc repeated measures ANOVA were performed to explore which feature(s) contributed to the pre- versus post-medication difference. An alpha-level of 0.05 was implemented and multiple comparison correction was performed using the false discovery rate (FDR) method described by Benjamini and Hochberg [45].

### 2.4. Classification of Medication States

#### 2.4.1. Individually Trained and Group Trained Models

Supervised classification analyses were performed to test whether differentiation between short-term pre- and post-medication was feasible, based on 60 s accelerometer features (Figure 1). First, this was tested using the four previously mentioned features extracted from the vector magnitude signal. Afterward, the feature set was expanded to include all described features, as well as the x, y, z time series (Appendix A). Analyses were performed using a support vector machine (SV) and a random forest (RF) classifier. Classification models trained on individual data and models trained on group data were then compared (Figure 1B).

For individually trained models, 80% of a participant’s total balanced data was used as training data, and 20% as test data (Appendix A). Small blocks (2%) of training data which neighbored the test data were discarded (Appendix A) to decrease the temporal dependence between training and test data. To prevent bias caused by the selected block of test data, a 41-fold cross-validation was performed. Each fold (out of 41) included two continual blocks of 10% of total data, one block from the pre- and one block from the post-medication recording as test data (percentiles 0 to 10 and 50–60, percentiles 1 to 11 and 51 to 61, …, and percentiles 40 to 50 and 90 to 100, see visualization in Appendix A).

For group-trained models, a leave one out cross-validation was performed. For every participant, a model was trained based on all data (balanced for medication status) from the remaining 19 participants and tested on all data (balanced for medication status) of the specific participant (Figure 1B). To assess all models, the area under the receiver operator curve (AUC) and the classification accuracy were calculated as predictive metrics. For the individual models, individual classification outcomes were averaged over the 41 folds. All parameters and details related to the implemented classifiers are available through the GitHub codebase [46].

To test the statistical significance of each individual and group model performance, 5000 permutation tests were performed in which medication state labels were shuffled. The 95th percentile of permutation scores was taken as significance threshold (alpha = 0.05), and FDR multiple comparison corrections were performed [45].

#### 2.4.2. The Influence of Training Data Size, and Feature Window Lengths

To test the impact of the size of the training set on the group models, the training phases were repeated with varying numbers of participants included in the training data. As in the original group model analysis, the test data consisted of all data from one participant. The number of training data participants varied between 1 and 19. To prevent selection bias in the selection of the training participants, the analyses were repeated five times per number of included training participants, with different random selections of training participants. Individual classification models were excluded from this analysis by definition.

To analyze the influence of feature window lengths, we repeated the group model analysis with features extracted from data windows of 3, 10, 30, 90, 120, 150, and 300 s duration. For every analysis, one participant was selected as a test participant, and the other 19 were training participants. This was repeated for all participants and the averages over 20 test participants were reported. This was performed at the group level modelling only, as individual models were limited by total available data size.

#### 2.4.3. Comparing Two Models’ Predictive Performance

Equality plots were drawn to compare the AUC scores and accuracies between two models, for example a model using 4 features versus 103 features, a model using an SV classifier versus an RF classifier, a model with versus without activity filtering (Appendix A). All comparisons were performed separately for the individual and group models. For example, model A led to a higher AUC score than model B in 14 out of 20 participants (14 dots above the equality line). Permutation tests plotted 20 random dots on an equality plot and tested whether the permuted distribution generated 14 or more dots (out of 20) above the equality line. This was repeated 5000 times, and the probability that the distribution 14 out of 20 was the result of chance was determined.

#### 2.4.4. Predictive Performance and Clinical Assessed Symptom Fluctuations

The influence of clinical bradykinesia, tremor, and abnormal involuntary movement fluctuations on predictive performance was tested at a group level by Spearman R correlations between the fluctuation in individual bradykinesia and tremor sub scores and AIMS scores, and the predictive performance (Appendix A). Individual participants were visualized according to descending tremor and AIMS fluctuation ratings to enable visual comparison of predictive performance with and without co-occurring tremor and abnormal involuntary movement fluctuation (respectively Appendix A). The tremor scores consisted of the MDS-UPDRS III items representing unilateral upper extremity tremor (items 15b, 16b, and 17b for the left side, and items 15a, 16a, and 17a for the right side).

#### 2.4.5. Software

Raw acceleration time series were down sampled and filtered (for gravity effects) in Matlab. All further preprocessing, feature extraction, and analysis was performed in Jupyter Notebook (Python 3.7). The code used to extract features and analyze data is made publically available [46].

#### 2.4.6. Code and Data Availability

The code used to extract features and analyze data is publicly available [46]. The de-identified open source dataset will be made available to the scientific community by the Michael J. Fox Foundation.

## 3. Results

### 3.1. Study Population and Recorded Data

Twenty PD patients from the Parkinson@Home data repository [37] were included in this study. We excluded three participants who did not show a levodopa-induced improvement in unilateral upper extremity bradykinesia and two participants were further excluded because there was less than 40 min of accelerometer data available from their pre- or post-medication recording, resulting in a dataset of 20 patients.

Demographic and disease-specific characteristics are presented in Table 1. In total, 3138 min of accelerometer data were recorded in the 20 included patients. After balancing the data sets for medication status, 2380 min of accelerometer data were included. On average, 59.5 (±14.3) min of accelerometer data from both pre- and post-medication recordings were included per participant. On average, 44.5 min (±13.9 min) of features were included after applying the activity filter and balancing the individual data to include equal individual features per medication state.

We extracted multiple features which are described in the current literature to index naturalistic bradykinesia with a wrist accelerometer (see Appendix A for details and references). In total, 103 motion accelerometer features were extracted for every feature window, including both time domain and spectral features from the accelerometer.

### 3.2. Group Level Statistical Analysis of Cardinal Motion Features across Medication States

At the group level, the pre- and post-medication recordings differed significantly regarding the individual means of the four main motion features (maximum acceleration magnitude, coefficient of variation of acceleration magnitude, root mean square of acceleration, and spectral power (below 4 Hz) [18,43]) (MANOVA, Wilk’s lamba = 0.389, F-value = 14.2, *p* < 0.001). Post-hoc repeated measures ANOVAs demonstrated that only the individual coefficient of variation averages significantly differed between pre- and post-medication states (*p* = 0.0042) (Figure 2).

### 3.3. Machine Learning Classification of Short Window Data Epochs

Next, the classification performance over short time windows (one-minute resolution) was tested using support vector (SV) and random forest (RF) machine learning models. All medication state classifications using the four previously selected motion features led to low AUC scores (means per model ranged between 0.49 and 0.64) and low accuracies (means per model ranged between 49% and 60%) (see Appendix A for detailed results).

We therefore repeated the classification analysis, using an expanded kinematic feature set (103 features). With the full feature set, notably higher AUC scores and classification accuracies were seen for all individual and group (SV and RF) models (Appendix A). Mean AUC scores per model ranged between 0.65 and 0.70, and mean accuracies per model ranged between 60% and 65% (Appendix A). Most participants yielded AUC scores and accuracies significantly better than chance level (17 out of 20 participants per model), tested through random surrogate dataset generation.

In 90% of participants (18 out of 20), either the best individual or group model classified medication states per 60 s significantly better than chance level based on our surrogate datasets (Figure 3 and Appendix A). Group trained models resulted in AUC scores statistically significantly higher than random classification in 17 participants. Individually trained models resulted in AUC scores statistically significantly higher than random classification in 13 participants. Both individual and group models resulted in mean AUC scores of 0.70 (±respectively 0.18 and 0.10), and mean accuracies of respectively 65% (±0.14) and 64% (±0.08) over all 20 participants (Figure 3, Appendix A). Notably, the individual models resulted in a larger standard deviation of AUC scores, including several AUC scores higher than 0.9 as well as below chance level (Figure 3).

Overall, these findings confirmed the feasibility of rapid naturalistic bradykinesia classification based on wrist-worn accelerometer metrics. Individual model training resulted in a similar mean AUC with a wider standard deviation compared to group model training.

### 3.4. Classification of Bradykinesia-Centred Motor Fluctuations Versus Co-Occurring Symptoms

We found no significant correlations between the individual classification performance and the individual clinically scored fluctuations in bradykinesia, tremor, and AIMS (see Appendix A, all *p*-values larger than 0.1). At an individual level, we found significant AUC scores in participants with (13, 24, and 79) and without (39, 51, and 58) tremor fluctuations (Appendix A). Similarly, we found significant AUC scores in participants with (2, 15, 51, and 79) and without (39, 18, 24, and 90) AIMS fluctuations (Appendix A).

Individual predictive performance was found not to be proportional to the size of tremor or AIMS fluctuations, suggesting the feasibility of using the applied metrics for PD patients with and without tremor and abnormal involuntary movements. Meanwhile, the severity of bradykinesia did not influence the classification performance, suggesting feasibility of the metrics for patients with even mild-to-moderate bradykinesia fluctuations.

### 3.5. Influence of Training Data Size and Feature Window Length

We found an increase in the predictive performance (AUC) of the group models as the number of patient datasets used during model training was increased (Figure 4A). Above 15 participants, the increase in mean AUC levelled off toward the 19 included participants.

Next, we wanted to investigate the impact of the accelerometer data feature window length on the predictive performance of the group models. Increasing the length of the feature windows up to 300 s improved the mean AUC (Figure 4B). Due to data size limitations, the feature windows were not expanded further than 300 s. These analyses could not be reproduced for the individual models due to data size limitations.

## 4. Discussion

Our results demonstrated successful classification of naturalistic bradykinesia fluctuations using wrist accelerometer data on different timescales using conventional statistical approaches (over one-hour epochs) and machine learning classification (over one-minute epochs). We found that the coefficient of variation of the accelerometer amplitude was significantly increased following dopaminergic medication when a full 60 min of data was analyzed per medication condition. At shorter timescales (60 s), this feature (complemented with 3 other accelerometer metrics) was not strongly predictive of medication state using machine learning. However, using a larger number of motion metrics (103), statistically significant classification of medication states could be achieved in 90% of participants (18 out of 20) using either group- or individually-trained models (Figure 3). Individual and group models both resulted in a mean AUC of 0.70 on the 60 s epochs, where the individual models’ AUC scores had a larger standard deviation (Figure 3, Appendix A). Expansion of the data epoch length (from 60 to 300 s), as well as inclusion of more training participants, improved AUC scores in the group models. Limited individual data sizes withheld us from testing individual models with expanded data epochs and may explain the larger standard deviation for individual model AUCs (Figure 1B and Appendix A).

These results represent the first demonstration of classification of Parkinsonian bradykinesia fluctuations using individually-trained models for single-wrist accelerometer data on a rapid timescale. Although we showed statistically significant classification over short time windows, we did not find added value yielded by individual model training based on our current results. Reproduction with longer accelerometer recordings for individuals is, however, likely to improve classification results further.

The significant difference of only the mean individual coefficient of variance between pre- and post-medication recordings (Figure 2) may be explained by a suggested larger discriminative potential of naturalistic sensor features describing a distribution rather than describing extreme values or sum scores [11].

In general, the presented classification models were notable due to the unconstrained naturalistic (real-world) data collection environment and short time scale of classification. Operating at this shorter timescale, the models showed good classification performance compared with benchmark naturalistic medication-state detection models (Figure 3) [34,35,36]. Although better classification performances have previously been described with models using data over longer timescales or from more constrained recording scenarios, these methodologies improved classification results at the cost of less naturalistic generalizability [18,22,27,28,43]. Adding a second motion sensor would likely increase performance at the cost of user-friendliness and feasibility [32]. Systems using wrist, ankle and/or axial motion sensors have the theoretical advantage of being more sensitive for arm versus leg- and gait-centered symptomatology.

### 4.1. Clinical Relevance and Methodological Challenges of Naturalistic and Rapid PD Motor Monitoring

Wearable accelerometer-based PD monitoring systems have been developed to augment therapeutic decision making, [20,21,23] and to augment clinical assessments in pharmacological trials [11,47]. Previous systems have been validated over the course of days. However, other suggested clinical state tracking applications would require short time scale feedback [22], including fine-grained cycle-by-cycle medication adjustments and conventional [48,49] or adaptive [50,51,52,53] deep brain stimulation programming. Moreover, the significant predictive performance in patients with and without both tremor and dyskinesia (AIMS) fluctuations underlines the potential of dynamic naturalistic monitoring for a wide spectrum of PD patients (Appendix A). The latter was important to investigate despite the filtering out of typical tremor bandwidths, since solely band pass filtering cannot rule out any influence of tremor dynamics.

Notably, we observed marked differences in classification accuracy using either 4 accelerometer features or 103 features (Figure 3 and Appendix A). This suggests that bradykinesia classification on shorter timescales, requires rich feature sets. The significance testing with surrogate datasets aimed to rule out any resulting overfitting. However, a thorough comparison of feature sets is often complicated by proprietary algorithms or the lack of open-source code [12]. This underlines the importance of transparent, open source, and reproducible movement metric feature sets for naturalistic PD monitoring [54,55].

Another methodological challenge for rapid, objective, naturalistic short-term PD monitoring is the lack of a high-quality labelling of data on the same time scale. PD clinical assessment tools, currently applied as gold standards, are limited in their applicability for rapid time scales. Multiple longitudinal time windows of the dynamic accelerometer time series are labelled with a single clinical score, which weakens model training and evaluation. In effect, sensor-based outcomes are often aggregated to match clinical evaluation metrics and time scales, which might account for the current upper limit in wearable classification performance [21,24,29]. PD-specific eDiaries [56,57,58,59] labelled video-recordings on fine time scales [60], and other virtual telemedicine concepts [61] may contribute to this challenge.

### 4.2. Future Scientific Opportunities to Improve Naturalistic PD Monitoring Development

We predict that the coming expansion of real-world motion data sets, containing long-term data over weeks to years in patients with PD, will support optimization of individually-trained models [62]. These larger datasets will also allow the exploration of alternative, more data-dependent, computational analyses, such as deep neural network classification and learning [35,63]. Moreover, unsupervised machine learning models could also be explored to overcome the issues of lacking temporally matching gold standard for model training and evaluation by surpassing the need for long-term, repetitive, true labels [11,64]. The observed discriminative potential of the coefficient of variation (Figure 2) might be of value in post hoc differentiation of clusters in unsupervised machine learning models.

Additionally, open-source research initiatives should catalyze the development of naturalistic PD monitor models which are not dependent on proprietary software [10,43]. The Mobilize-D consortium, for example, introduced a roadmap to standardize and structure naturalistic PD monitoring by creating specific “unified digital mobility outcomes” [54,55]. During the development of these outcomes, features describing distribution ranges and extreme values—rather than means or medians—should be considered [11]. Parallel to open-source initiatives, other creative collaborations between industry and academia, such as data challenges, might offer valuable (interdisciplinary) cross-fertilization [65]. Furthermore, adding more limb sensors to improve naturalistic PD monitoring is controversial. Although there is evidence supporting the combined use of wrist, ankle [32,66], or insoles [67] sensory tracking, other reports have not shown improved performance and instead described additional burden to the patient [35,68,69].

### 4.3. Limitations

Our study was limited by the individual data set sizes, which restricted inferences that could be made regarding models trained with individual versus group data. Additionally, the unconstrained character of the pre- and post-medication recordings led to an imbalance in terms of captured activities during the two medication states. The applied activity-filter addressed this limitation partly but does not rule out imbalance in exact activities. This imbalance compromises pattern recognition based data analysis [35], but is also inherent to naturalistic PD monitoring [18]. Exploring the boundaries of this limitation is essential for future PD monitor applications. Replication of our methodologies in larger data sets, and inclusion of validated activity classifiers may contribute to overcoming this limitation. Future studies should also aim to detect symptom states beyond a binary differentiation between on- versus off-medication.

## 5. Conclusions

We demonstrated that classification of naturalistic bradykinesia fluctuations at the single-minute time scale was feasible with machine learning models trained on both individual and group data in PD patients using a single wrist-worn accelerometer. At longer timescales (i.e., an hour), a single accelerometer feature, the coefficient of variation, was predictive of bradykinesia at the group level. Extension of short accelerometer time epochs and an increased number of training patients improved classification of group-trained models. Rapid, dynamic monitoring has the potential to support personalized and precise therapeutic optimization with medication and stimulation therapies in Parkinson’s patients.

## Figures and Tables

**Figure 1 sensors-21-07876-f001:**
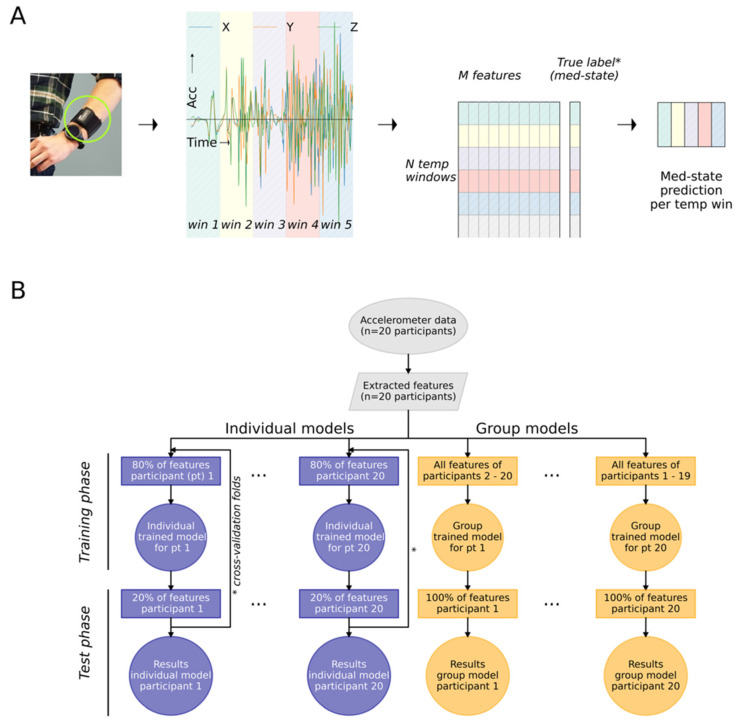
Accelerometer-based Parkinsonian motor fluctuation detection workflows. (**A**) Left: A wrist-worn motion sensor (Physilog 4, Gait Up SA, CH; green circled) is used to collect unilateral tri-axial accelerometer data. X, Y, and Z represent acceleration (meters/second per second) in three axes over time (seconds). Temporal windows are determined for data analysis and are indicated in different colors over time (win1, win2, …). Center: Signal preprocessing and feature extraction convert the raw tri-axial signal into a dataset containing M features (Appendix A), calculated for every temporal window (in total M columns and N rows). For the training phase of the machine learning classification models, the true labels representing medication states (*) are used. Right: In the testing phase, inserting the feature set (M × N) in the trained classification model leads to N medication state predictions. (**B**) Workflow to train and test individual and group models. Identical features are extracted from the raw accelerometer data of the twenty included participants (grey symbols). For the individually trained models (blue), the features from 80% of a participant’s epochs are used in the *training phase* (*y*-axis). The trained individual model is tested with the remaining, unused, 20% of epochs during the *test phase*. The arrows (*) from test phase to training phase represent the multiple cross-validation folds applied to train and test the individual models on different selections of training and test data. For the group models (yellow), each participant is tested in turn, with data from the other 19 participants used in the *training phase*.

**Figure 2 sensors-21-07876-f002:**
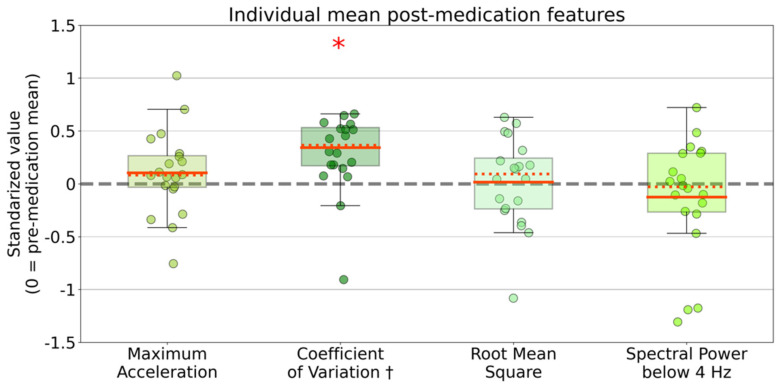
Distributions of individual means for four main movement features. Colored dots represent the mean feature values during the whole post-medication recording per participant (*n* = 20). Individual post-medication mean values are standardized as z-scores (individual pre-medication recordings are used as references, and therefore the pre-medication mean values equal 0). The red asterisk indicates a significant difference on group level between mean coefficient of variations of pre- and post-medication means (alpha = 0.05, MANOVA and post-hoc analysis, FDR corrected). † = one positive outlier (1.7) not visualized.

**Figure 3 sensors-21-07876-f003:**
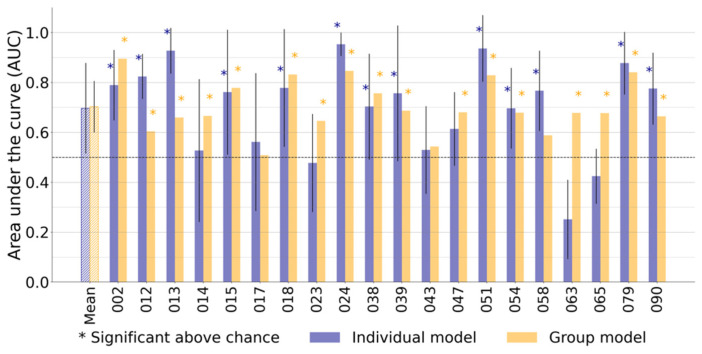
Classification of medication induced motor fluctuations on short accelerometer time windows in individual participants. The first pair of bars represents the mean area under the curve (AUC) score over the twenty participants. Each subsequent pair of bars (002 to 090) represents the AUC scores from one participant. The blue bars represent the AUC score for the individual model, and the yellow bars represent the group model. Note that for the individual models, AUC scores are the averages over the multiple cross-validation folds within a participant (Figure 1B). The asterisks indicate whether the corresponding AUC score was significantly better than chance level (5000-repetitions permutation test). Both models have equal mean AUC scores. It is notable that the majority (18 out of 20 of participants) has at least one significant score. Half of the participants yielded a higher AUC score with the individual model than with the group model.

**Figure 4 sensors-21-07876-f004:**
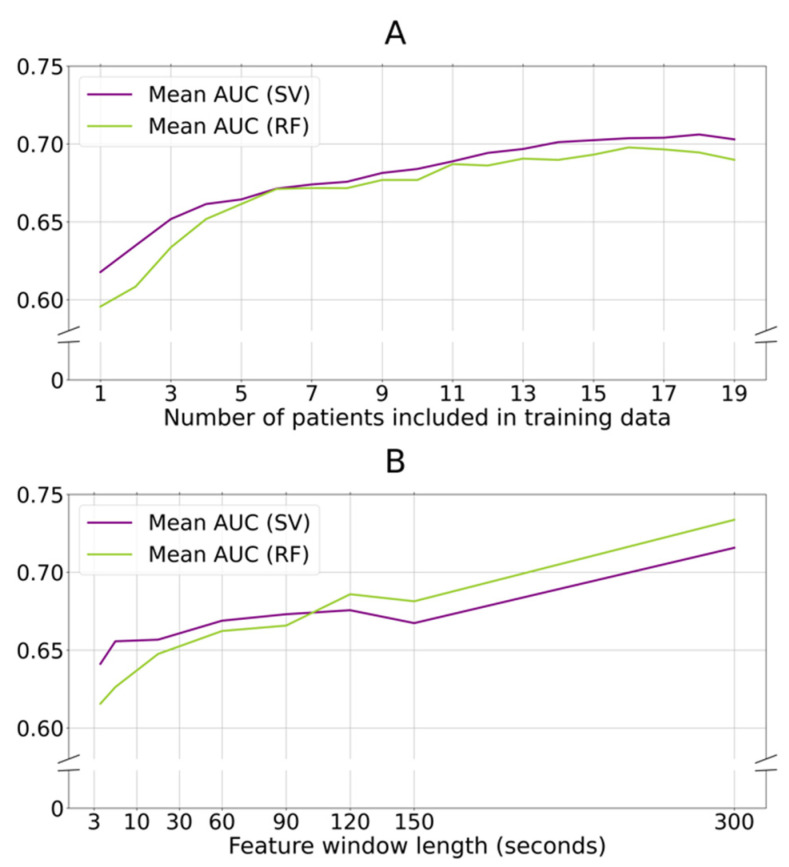
Increasing number of training patients and length of data window duration improves classification performance. (**A**): Group models are trained for every patient with a varying number of included training data, (*x*-axis). On the *y*-axis, the AUC is shown for both SV and RF models (both included activity filtering). An increase in AUC is seen for SV and RF models parallel to an increase in included training patients. (**B**): Group models are trained for every patient with various feature window lengths (*x*-axis). On the *y*-axis, the AUC of the SV and RF models are visualized. Due to the longer feature window lengths, the activity filter in this sub-analysis is not applied to data size limitations. Larger window lengths up to 300 s increase classification performance, while smaller window lengths decrease classification performance. AUC: area under the receiver operator characteristic; SV: support vector classifier; RF: random forest classifier.

**Table 1 sensors-21-07876-t001:** Demographic and disease specific characteristics of patient population. AIMS: Abnormal Involuntary Movement Scale. MDS-UPDRS: Movement Disorders Society Unified Parkinson Disease Rating Scale. Sd: standard deviation.

*Characteristics*	
Total number (% female)	20 (60%)
Age (years, mean (sd))	63.4 (6.4)
Accelerometer data per medication state (minutes, mean (sd))	59.5 (14.3)
Accelerometer data per medication state, after activity filtering (minutes, mean (sd))	44.5 (13.9)
PD duration (years, mean (sd))	8.1 (3.5)
Levodopa equivalent daily dosage (milligrams, mean (sd))	959 (314)
MDS-UPDRS III pre-medication	43.8 (11.6)
MDS-UPDRS III post-medication	27.1 (9.6)
AIMS pre-medication	0.5 (1.8)
AIMS post-medication	3.7 (4.2)

## Data Availability

The de-identified open source dataset will be made available to the scientific community by the Michael J. Fox Foundation.

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
