# Peer review of "Rapid Dynamic Naturalistic Monitoring of Bradykinesia in Parkinson’s Disease Using a Wrist-Worn Accelerometer"

_sensors, 2021, doi:10.3390/s21237876_

Round 1

Reviewer 1 Report

This paper describes the examination of different machine learning algorithms to classify on- and off-medication states of Parkinson’s disease patients in a naturalistic environment.

The idea is good, and I think that having a tool that can detect motor fluctuations with high time resolution has a practical value for more precise patient monitoring. However, specific points in the paper need to be clarified. The comments are listed below.

  1. Section 2.2 said that data were low-pass filtered at 3.5 Hz before further preprocessing. However, later in the text, it is mentioned that one feature represents the total spectral power below 4 Hz. Why were different cut-off frequencies used?
  2. I would recommend that the total number of included patients is mentioned in Section 2.1., since it is later referred in the text. It is explicitly mentioned only later in the Results section, and it can be a bit confusing.
  3. In the same section, line 125, it is mentioned that only data from the side with the largest clinical change in upper extremity sub items were included. Does this clinical change represent the difference between the on- and off-states sum scores? I would suggest explaining this more intuitively.
  4. In the Methods section, line 232, it is mentioned that the threshold is selected based on some video-annotated sections identified as non-active. Were those sections included in the used database? If so, it should be mentioned in the Methods section that the data was also recorded with a video camera (even though you referenced your previous work explaining this). How many sections were used for threshold determination?
  5. The total number of extracted features should be mentioned in Section 2.2. for the same reasons as those in point 2.
  6. Some details about the implemented classifiers should be included in the text, e.g., the choice of kernel, number of training epochs, etc.
  7. Did you maybe examine the possibility of data fluctuations during the one hour of recordings? One hour of activities may induce fatigue in patients’ performance and therefore influence the classification outcome.
  8. It seems that some features are correlated. A simpler model could be trained by excluding redundant features, and therefore, better results could be obtained. Have you tested some feature selection methods since you have a large number of features?
  9. Figure 1 should be organized in a better way, e.g,. by mentioning the y-axis for the training phase, it can be confused with the accelerometer y-axis. The mark * is used in two different places, in both A and B panels.
  10. Activity filtering should be described before classification since it represents the processing step that precedes the classification.
  11. Why did you examine the influence of the number of training data participants since it is expected that more data would give better results? It seems a bit redundant.
  12. Table 1 is referenced earlier in the text (line 164), but the Table is shown later in the paper. It should be moved below the text where it is first mentioned. Furthermore, the Table shows the characteristics of accelerometer data that are not demographic or disease-specific. Also, the legend contains some marks that are not present in the Table (* and **).
  13. The supplementary data should be organized in the order in which it is mentioned in the text.
  14. In Figure S2, the x-axis should be labeled as a time axis.
  15. Would you please check the typos in the text (e.g., line 52)? Furthermore, the equations should be written with the equation editor (e.g., line 146, line 295).
  16. Why did you try to find correlations between the classification performance and tremor scores since data was previously low pass filtered to remove tremor-related frequencies (line 144)? In that sense, it seems incorrect to conclude that the applied metrics is feasible for PD patients with and without tremor.
  17. In your opinion, what is the minimum length of data that would provide better classification results but also give predictions on a rapid time scale?
  18. I agree that imbalance in exact activities during different medication states represents a limitation that can influence the results. It would be interesting to see this explored as part of future work. Since you have video recordings of these daily activities, maybe you could explore if some activities provided better results, representing an interesting basis for future work.

Reviewer 2 Report

In this study, the authors tested the influence of individual model training on short timescale classification of naturalistic (at home) bradykinesia fluctuations in patients with Parkinson’s disease (PD). For this aim, they used data derived from a wrist accelerometer. They compared classification models based on individual data with classification models based on group data. They also analysed the influence of several factors, including the number of individuals and length of analysed data epochs, on classification results.  The analysis showed that at long timescales (60 minutes of data analysed for medication condition), the coefficient of variation of the accelerometer amplitude significantly increased after dopaminergic medication. When considering shorter timescales (60 seconds), this kinematic feature was not predictive of medication state using machine learning. However, considering a larger number of motion metrics, statistically significant classification of medication states could be achieved in 90% of patients using either group or individually trained models. The area under the curves (AUCs) of the group models improved as the length of the feature windows was increased and with additional training patient datasets. When discussing their results, the authors emphasize the feasibility of classifying bradykinesia fluctuations in PD patients using individually trained models for single wrist-accelerometer data on a rapid timescale. This has potential therapeutic implications.  

The present study discusses an interesting topic and it is well-designed. The paper is sufficiently well written; the study aims are clear. The methods and the results are well described.  Study limitations are extensively discussed. 

I have only one minor comment:
To test the statistical distinguishability of the pre- and post-medication recordings at the group level, the authors selected a priori 4 accelerometer features from the entire dataset, i.e., maximum acceleration, coefficient of variation of acceleration over time, root mean square of acceleration over time, and the total spectral power below 4 Hz. This methodological approach was justified by the observation that these 4 features covered the most often used domains of motion metrics applied for naturalistic bradykinesia monitoring. However, only the coefficient of variation differed between pre- and post- medications states after post hoc comparisons. Could the authors further discuss this result? What the coefficient of variation may represent as compared to the other three accelerometer features?

Author Response

Reviewer 2

In this study, the authors tested the influence of individual model training on short timescale classification of naturalistic (at home) bradykinesia fluctuations in patients with Parkinson’s disease (PD). For this aim, they used data derived from a wrist accelerometer. They compared classification models based on individual data with classification models based on group data. They also analysed the influence of several factors, including the number of individuals and length of analysed data epochs, on classification results.  The analysis showed that at long timescales (60 minutes of data analysed for medication condition), the coefficient of variation of the accelerometer amplitude significantly increased after dopaminergic medication. When considering shorter timescales (60 seconds), this kinematic feature was not predictive of medication state using machine learning. However, considering a larger number of motion metrics, statistically significant classification of medication states could be achieved in 90% of patients using either group or individually trained models. The area under the curves (AUCs) of the group models improved as the length of the feature windows was increased and with additional training patient datasets. When discussing their results, the authors emphasize the feasibility of classifying bradykinesia fluctuations in PD patients using individually trained models for single wrist-accelerometer data on a rapid timescale. This has potential therapeutic implications.  

The present study discusses an interesting topic and it is well-designed. The paper is sufficiently well written; the study aims are clear. The methods and the results are well described.  Study limitations are extensively discussed. 

We thank the reviewer for the excellent summary and supportive evaluation. 

I have only one minor comment:

To test the statistical distinguishability of the pre- and post-medication recordings at the group level, the authors selected a priori 4 accelerometer features from the entire dataset, i.e., maximum acceleration, coefficient of variation of acceleration over time, root mean square of acceleration over time, and the total spectral power below 4 Hz. This methodological approach was justified by the observation that these 4 features covered the most often used domains of motion metrics applied for naturalistic bradykinesia monitoring. However, only the coefficient of variation differed between pre- and post- medications states after post hoc comparisons. Could the authors further discuss this result? What the coefficient of variation may represent as compared to the other three accelerometer features?

Yes - indeed this was an interesting finding to us. Contrarily to the other three features, the coefficient of variance describes a summary metric for the distribution of the motion, instead of simply a maximum or sum score. We suggest that longer term features for naturalistic sensor data in Parkinson patients may have more discriminative potential when they represent the distribution of acceleration. This suggestion is supported in earlier work of Warmerdam et al. We have now added this component to the discussion (p. 11, l. 420-423).